# Comprehensive Study of Components and Antimicrobial Properties of Essential Oil Extracted from *Carum carvi* L. Seeds

**DOI:** 10.3390/antibiotics12030591

**Published:** 2023-03-16

**Authors:** Changhai Liu, Feng Cheng, Haji Akber Aisa, Maitinuer Maiwulanjiang

**Affiliations:** 1Xinjiang Key Laboratory of Plant Resources and Natural Products Chemistry, Xinjiang Technical Institute of Physics and Chemistry, Chinese Academy of Sciences, Urumqi 830011, China; 2College of Chemical Sciences, University of the Chinese Academy of Sciences, Beijing 100039, China

**Keywords:** *Carum carvi* L., essential oil, MRSA, GC-Q-TOF-MS, metabolomics

## Abstract

*Carum carvi* L. belongs to the *Apiaceae* family and is widely used as a vegetable, food spice, preservative, and herbal medicine. This study investigated the impact of essential oil extracted from *Carum carvi* L. seeds (CEO) on methicillin-resistant *Staphylococcus aureus* (MRSA) and its possible action mechanism. The dominant chemical components of CEO determined by GC-MS were carvone and limonene. It was observed that CEO had a considerable inhibitory effect against the growth of planktonic bacteria and biofilm in MRSA cells. Untargeted metabolomics based on GC-Q-TOF-MS was used to analyze the possible mechanism of the interaction of MRSA with CEO. It was determined that there were 63 different metabolites based on fold change values greater than 1.5 or less than 1.5, *p* < 0.05, VIP > 1, which demonstrated amino acid metabolism in MRSA was significantly affected by CEO. In conclusion, CEO has a potent antimicrobial property and has promising potential for use in food and drugs.

## 1. Introduction

As a typical Gram-positive bacterium, *Staphylococcus aureus* (*S. aureus*) has been identified as one of the major pathogens that pose threats to human health [1,2,3]. Normally, *S. aureus* cannot cause any disease, but in some cases, it can cause an extensive variety of infections, for instance, hospital-acquired infections, community-acquired infections, pneumonia, sepsis and skin infections, etc. [4]. In serious cases, *S. aureus* infections can lead to fatal bacteremia and sepsis [5]. Due to the irrational use of antibiotics and antimicrobials in recent years, drug-resistant *S. aureus* has been widely disseminated, especially with the emergence of MRSA (methicillin-resistant *Staphylococcus aureus*), which has led to public panic about the problem of bacterial resistance [6]. MRSA, as a notorious pathogen, is highly resistant to β-lactams and many other antibiotics, making the infections more difficult to treat [7,8]. Furthermore, MRSA can also easily develop biofilms, which can lead to a substantial increase in its drug resistance and virulence [9,10]. It has been reported that the sensitivity of planktonic bacteria to antibiotics is approximately 10–1000 times lower than that of biofilms [11]. Therefore, there is an urgent need to develop novel drugs to control MRSA.

In recent decades, many natural products of plant origin, such as essential oils, have gained considerable attention as potential new generations of antibiotics [12,13]. Essential oils (EOs) are volatile secondary metabolites of plants that can protect plants from environmental and pathogenic microorganisms and are mainly composed of phenols, monoterpenes, sesquiterpenes, and other aromatic compounds. Due to their unique aroma and flavor, EOs have been widely applied in biomedicine, pharmaceuticals, cosmetics, food, agriculture, and other fields [14]. To date, more than 3000 EOs have been identified by researchers, but only about 300 of them have been put to practical use [15].

*Carum carvi* L. (*C. carvi*, Caraway), belonging to the *Apiaceae* family, has been widely used as a vegetable, food spice, preservative, and herbal folk medicine. It is an important medicinal plant from the *Apiaceae* family and has been cultivated for a long time in the north and center of Europe, Egypt, Australia, Iran, and China [16]. Because of its unique flavor, Caraway is often used as a spice in food, while it is also mainly employed in medicine for liver protection, deworming, diuresis, and antioxidation for its variety of biological activities [16]. Studies have demonstrated that caraway essential oil (CEO) possesses broad biological activities [17,18] such as anti-inflammatory [19] and antioxidant [17]. Furthermore, CEO has been found to have an excellent inhibitory effect on various microorganisms, including *Clavibacter*, *Agrobacterium* [18], *Sitophilus oryzae* L. [20], *Aspergillus flavus* [21], and *Vibrio* spp. [22]. However, early data were only focused on the antibacterial activity, and very limited studies reported its antibacterial action mechanism. Therefore, this study investigated the inhibitory activity of CEO against MRSA and its biofilm and determined the potential mechanism by untargeted metabolomics based on GC-Q-TOF-MS for the first time. This research provides a reference for the research and development of traditional essential oils.

## 2. Materials and Methods

### 2.1. Materials

*Carum carvi* L. was collected from Qinghai, China. A voucher specimen of the plant sample was deposited in the herbarium of the Xinjiang Technical Institute of Physics and Chemistry, Chinese Academy of Science (voucher specimen no. WY02212). All the culture media in this study were obtained from Hopebiol (Qingdao, China). Bis-trimethylsilyltrifluoroacetamide (BSTFA) was obtained from Beijing Solarbio Science & Technology (Beijing, China). Pyridine was obtained from Aladdin (Shanghai, China). Adontiol was obtained from Sigma Aldrich (Shanghai, China).

### 2.2. Bacterial Strains and Media

MRSA (ATCC 43300) and *E. coli* (ATCC 43895) used in this study were obtained from Beina Biotechnology (Beijing, China), while *S. aureus* (ATCC 6538) and another *E. coli* (ATCC 25922) was preserved in the laboratory. The strain stored at −80 °C was revitalized in Mueller–Hinton Broth (MHB) at 37 °C for 16 h with shaking incubation (180 rpm).

### 2.3. Preparation of CEO and Component Analysis

The CEO was extracted by hydrodistillation in a Clevenger apparatus for 3 h from the seeds of *C. carvi*. The CEO was diluted 1:20 (*v*/*v*) in *n*-hexane for component analysis. The components of CEO were analyzed by an Agilent 7200 Series Q-TOF GC/MS System (Agilent Technologies, Inc., Santa Clara, CA, USA) with an HP-5MS column (30 m × 250 μm × 0.25 μm). The flow rate of the carrier gas, helium, was 1.0 mL/min, while the temperature of the injector was 280 °C. The initial oven temperature was 60 °C and maintained for 5 min; then, the oven temperature was gradually raised to 280 °C at 4 °C/min and was finally maintained for 2 min. A 0.3 μL sample was injected in split mode (30:1). The mass spectrometer was operated in the range of 50−500 *m*/*z*, and electron impact ionization (EI) was set as 70 eV. The retention index (RI) was calculated by the retention time of *n*-alkanes. The components of CEO were identified by comparing with RI and the National Institute of Standards and Technology 14 (NIST 14) library.

### 2.4. Testing the Susceptibility of Planktonic Bacteria

The MIC (minimum inhibitory concentration) value of CEO against bacteria was determined according to the Clinical and Laboratory Standard Institute (CLSI). The MIC was considered to be the lowest concentration of the CEO that could inhibit the growth of bacteria. To test the MBC (minimum bactericidal concentration), 100 μL of culture solution in the well without bacteria growth was cultured on MHA and incubated at 37 °C for 16 h. The lowest concentration without bacteria growth on the MHA was considered to be the MBC of CEO against bacteria.

### 2.5. Time Kill Curve

Overnight cultured MRSA was inoculated into MHB containing CEO at different concentrations (0.08%, 0.5 × MIC; 0.16%, 1 × MIC; 0.32%, 2 × MIC) and cultured at 37 °C. After a specific time interval, the OD_600_ of the culture solution was determined by a microplate reader (BIO-RAD, Contra Costa, CA, USA). Every sample was tested in triplicate.

### 2.6. The Effect on Biofilm

The effect on biofilm of MRSA was measured by crystal violet as in the previous report [23]. The inhibition and clearance of MRSA biofilm were determined. To determine the impact of CEO on the removal of biofilms, firstly, biofilm was cultured in TSB-g (TSB with 1% glucose) for 24 h. Then different concentrations of CEO (1.28%, 0.64%, 0.32%, 0.16%, 0.08%, and 0) were added. After a 24 h period, the culture medium from the 96-well plate was disposed of, and each well was washed three times with PBS. After drying, each well was stained with 150 μL of crystal violet solution for 15 min. Then, the remaining crystal violet was rinsed with tap water and the microporous plate was dried. Finally, 95% ethanol was added and re-dissolved for 15 min, and then, the optical density at 600 nm (OD600) was determined. The amount of biofilm was determined by the amount of crystal violet dissolved in 95% ethanol. For the inhibition of biofilm formation, CEO was diluted in TSB-g medium (1.28%, 0.64%, 0.32%, 0.16%, 0.08%, and 0); then, MRSA was added and cultured for various time intervals to observe the inhibition of MRSA biofilm formation. Each experiment was replicated thrice.
$$Clearance\;rate\left(\%\right)=\frac{OD_{control}-OD_{treatment}}{OD_{control}}\times 100\%$$

### 2.7. Metabolite Extraction and Analysis

Cultured MRSA was inoculated into MHB at 37 °C for 12 h with shaking incubation (180 rpm), and then, samples were exposed to CEO at a concentration of 0.08% for 6 h. Bacteria were collected by centrifugation at 12,000 rpm for 5 min, followed by being washed three times using PBS. After weighing the samples, 50% methanol in water was used to extract metabolites. The supernatant of each sample was collected, and this was followed by adding 10 μL of 0.1 mg/mL adonitol as the internal standard. All samples were dried by a nitrogen blower (Organomation, Berlin, MA, USA). The derivatization of the samples was performed with a bit of adjustment based on the previous report [24]. Specifically, 100 μL of 20 mg/mL methoxyamine hydrochloride pyridine solution was used to protect carbonyl through a 60 min 37 °C reaction. BSTFA containing 1% trimethylsilyl (TMCS) was used to derivatize samples through a 90 min 70 °C reaction.

GC-MS analysis was determined by an Agilent 7200 Series Q-TOF GC/MS System with an HP-5MS column (30 m × 250 μm × 0.25 μm). The initial oven temperature was 60 °C and was maintained for 3 min; then, the oven temperature was gradually raised to 280 °C at 5 °C/min and was finally maintained for 1 min. Helium was used as the carrier gas at a flow rate of 1 mL/min. The mass spectrometer was operated in the range of 33–600 *m*/*z*, and EI energy was set as 70 eV. The injection volume was set as 1 μL in the splitless mode. The data collection type was centroid. Each sample was tested with six replicates.

MS-DIAL (version 4.9.0) was used to analyze raw data, including data collection, peak detection, noise reduction, alignment, and identification. MassBank NIST was used to identify metabolites [25]. The processed data were first analyzed by the principal component analysis (PCA) by MetaboAnalyst (version 5.0, accessed on 26 October 2022, https://www.metaboanalyst.ca/) to evaluate differences and similarities between the samples. A t-test and orthogonal partial least squares discriminant analysis (OPLS-DA) were performed to obtain the differential metabolites by MetaboAnalyst [26]. Metabolites with VIP > 1, fold change > 1.5, or fold change < 0.67 were considered to be differential metabolites. Kyoto Encyclopedia of Genes and Genomes (KEGG) enrichment analysis performed by MetaboAnalyst was used to analyze pathways that the differential metabolites were involved in [27].

### 2.8. Statistical Analysis

All data were reported as means ± SD. The significance of differences was determined by a *t*-test using SPSS 23.0 (SPSS Inc., Chicago, IL, USA). A value of *p* < 0.05 was considered to be significant.

## 3. Results

### 3.1. Chemical Component of CEO

The analysis of the dominant chemical components of CEO, as presented in Table 1, was conducted using GC–MS. There were 10 components identified, representing 99.15% of the total CEO. The major constituent of CEO was carvone (69.7%), followed by limonene (28.55%).

### 3.2. Antibacterial Activity

The MIC and MBC values, as presented in Table 2, were used to evaluate the inhibitory impact of CEO on the growth of bacteria. The result indicated that CEO exhibited good inhibitory activity against bacteria. As shown in Figure 1, a time–kill curve of MRSA exposed to CEO was established. It was observed that the growth of MRSA occurred upon exposure to CEO at concentrations of 1 × MIC (0.16%, *v*/*v*) and 2 × MIC (0.32%, *v*/*v*). Even at the sub-inhibitory concentration (0.08%, *v*/*v*), a reduction of 20% in the number of MRSA was observed when compared to the control group, suggesting that CEO possesses bactericidal properties. Furthermore, it also could be seen that CEO could inhibit the growth of MRSA in a concentration-dependent manner.

### 3.3. Antibiofilm Activity

Since forming biofilm is one of the crucial roles for MRSA to produce drug resistance, the impact of CEO on mature biofilm and biofilm formation of MRSA was assessed using crystal violet staining. As shown in Figure 2A, CEO at various concentrations displayed distinct antibiofilm activity and demonstrated a dose-dependent response. CEO at a concentration of 1.28% was able to completely inhibit the formation of MRSA biofilm, and with the decrease in concentration, the inhibition effect diminished progressively. When the concentration of CEO came to 0.08% or 0.16%, it exhibited minimal inhibitory activity on biofilm formation.

The destruction of the CEO on mature cultured biofilm was conducted to evaluate the effect on mature biofilm of MRSA. As depicted in Figure 2B, the clearance rate of biofilm biomasses increased from 0.085% to 88.60% with concentrations of CEO increasing from 0.08% to 1.28%. CEO at a concentration of 2 × MIC (0.64%, *v*/*v*) or above could significantly reduce the biofilm biomasses compared to the control group (*p* < 0.05), indicating the destructive effect was dose-dependent.

### 3.4. Metabolomics

In this study, the impact of EO on the metabolism of MRSA was investigated based on GC-Q-TOF-MS. As shown in Figure 3C, the PCA analysis of twelve sets of data from the six biological replicates revealed that the replicates in each group were clustered, while the two groups were well separated. After data processing, there were 213 metabolites identified.

In order to identify the changes in metabolites that occurred in MRSA between the CEO-treated group and the control group (without CEO-treated), Orthogon Partial Least Squares Discriminant Analysis (OPLSDA) was employed to screen differential metabolites (Figure 4A). The Q^2^ of this model was 0.898, showing that there was a high degree of validated predictability, and it could be utilized for further analysis of the screening of differential metabolites. A Student’s *t*-test and fold change (FC) were employed to calculate significant differences between the two groups. The metabolites with FC > 1.5 or FC < 0.67, *p* < 0.05, and VIP value > 1.0 were considered to be differential metabolites (Figure 4B). A total of 64 differential metabolites were screened, including 4 upregulated and 60 downregulated in the CEO-treated group relative to the control group. The details of metabolite changes, including VIP values, fold changes, and *p* values, are presented in Appendix A.

As shown in Figure 4C, KEGG enrichment analysis was conducted based on the differential metabolites screened in the preceding step. A total of 26 metabolic pathways were identified, and 10 were significantly enriched (*p* < 0.05), including aminoacyl-tRNA biosynthesis, valine, leucine, and isoleucine biosynthesis, and metabolism of amino acids, such as alanine, aspartate, and glutamate. Among the differential metabolites and pathways which were involved, it was observed that many amino acids were key metabolites that played a pivotal role after MRSA’s exposure to CEO. The abundance changes between the two groups are depicted in Figure 4D.

## 4. Discussions

The main chemical components of CEO found in this study were limonene (28.55%) and carvone (69.7%), which accounted for more than 98% of CEO. This result is different from the previous research [22] and European Pharmacopoeia, which also defines that CEO should contain limonene (30~45%), carvone (50~65%), and *β*-myrcene. Still, it is similar to the CEO from German caraway containing carvone (77.3%) and limonene (16.2%) [28]. In brief, the CEO in Qinghai is similar to that in other places, but it is a bit different due to its higher content of limonene and carvone.

Previous research [17,29] showed that CEO has antimicrobial activity and also has influence on the biofilm formed by *Pseudomonas aeruginosa*. However, there are few studies about its activity and mechanism against MRSA. Herein, we focused on the antibacterial activity of CEO. For most bacteria in the test, the MBC was two times or four times the MIC, indicating that CEO exhibits bactericidal properties [30]. In addition, the time–kill curve of CEO against MRSA also suggested that CEO showed good bactericidal activity in a concentration-dependent manner. Studies have demonstrated that CEO has anti-quorum sensing activity against *Gram*-negative bacteria, which could inhibit the formation of biofilm of *Vibrio*, *Klebsiella*, and *Pseudomonas aeruginosa* [22,31]. The influence on the biofilm of MRSA confirmed that CEO showed great antibiofilm activity against *Gram*-positive bacteria; however, the underlying mechanism requires further investigation.

In order to investigate the mechanism of CEO against MRSA, metabolomics based on GC-MS was performed. It was observed that the metabolism of MRSA was significantly impacted in comparison to the control group, with 64 differential metabolites and 26 metabolic pathways being identified. Furthermore, a decrease in the content of most amino acids was observed, suggesting that CEO shows a substantial effect on the metabolism of amino acids of MRSA. The metabolism of amino acids is a crucial pathway for the growth and survival activities of MRSA, as it is involved in protein synthesis, nutrient uptake, and biofilm formation [32,33]. Branched-chain amino acids (BCCAs), including valine, leucine, and isoleucine, are essential nutrients in *S. aureus* as they are necessary for the synthesis of protein and membrane branched-chain fatty acids (BCFAs) [34], which are the vital part of the cell membrane and are essential for virulence and maintaining the fluidity of the cell membrane [35]. The destruction of cell membrane integrity will ultimately result in the leakage of cell contents, such as DNA, proteins, and amino acids. In our study, the decrease in the content of multiple amino acids may be attributed to the disruption of the integrity of MRSA biofilm, with the loss of BCAAs further exacerbating this process. Similarly, many other studies have demonstrated that essential oils can have an impact on the integrity of bacterial cell membranes [36,37,38].

Pyrimidine is a crucial raw material for the synthesis of DNA and RNA. In this study, some metabolites associated with pyrimidine were significantly decreased, such as uridine, thymine, and uracil, which led to obstacles in synthesizing DNA and RNA. Additionally, pyrimidine has been shown to have an inhibitory effect on biofilm formation by suppressing the production of signal molecules and extracellular DNA (eDNA) [39,40]. Previous studies have reported similar findings. Terpinen-4-ol, the primary constituent of tea tree oil, has been found to destroy MRSA biofilm by inhibiting the biosynthesis of pyrimidine nucleotides [41].

## 5. Conclusions

In this study, it was demonstrated that CEO possesses strong antibacterial activity and has a significant effect on MRSA biofilm formation. Furthermore, untargeted metabolomics based on GC-Q-TOF-MS was employed to analyze the potential mechanism of the response of MRSA exposed to CEO. It demonstrated that CEO disrupts the integrity of the MRSA cell membrane, resulting in the efflux of its contents, such as amino acids, and thus exerting its antibacterial activity. To the best of our knowledge, this is the first report on the mechanism of CEO against MRSA, suggesting that CEO could be a promising natural antibacterial product for addressing the increasing prevalence of drug-resistant bacteria.

## Figures and Tables

**Figure 1 antibiotics-12-00591-f001:**
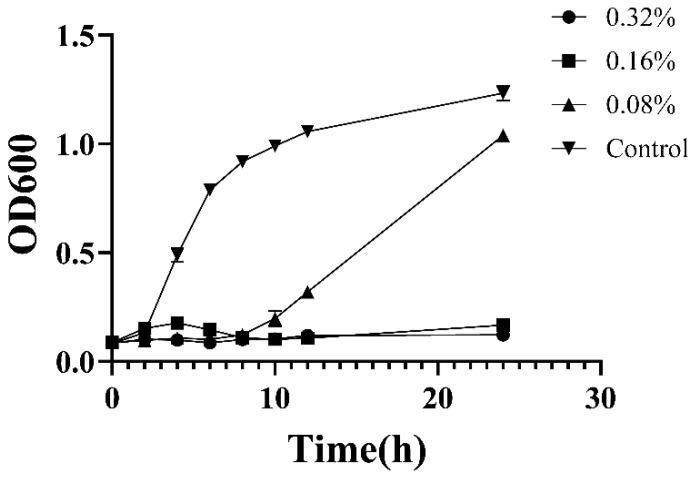
Time–kill curve of MRSA treatment with CEO.

**Figure 2 antibiotics-12-00591-f002:**
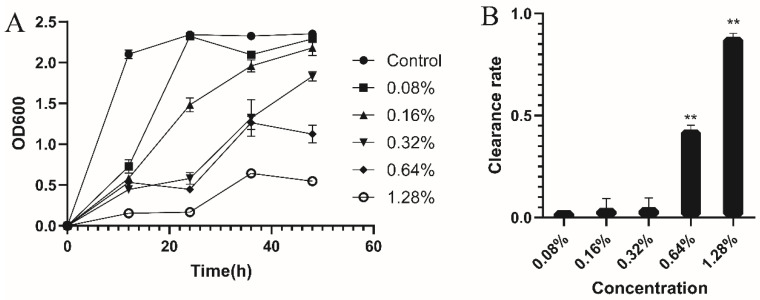
Impact of CEO on MRSA biofilm. (**A**): Effect of CEO on the formation of MRSA biofilms; the ordinate represents the relative content of MRSA biofilms. (**B**): Disruption of mature biofilms; the ordinate represents the clearance rate of MRSA biofilm by CEO. ** *p* < 0.05.

**Figure 3 antibiotics-12-00591-f003:**
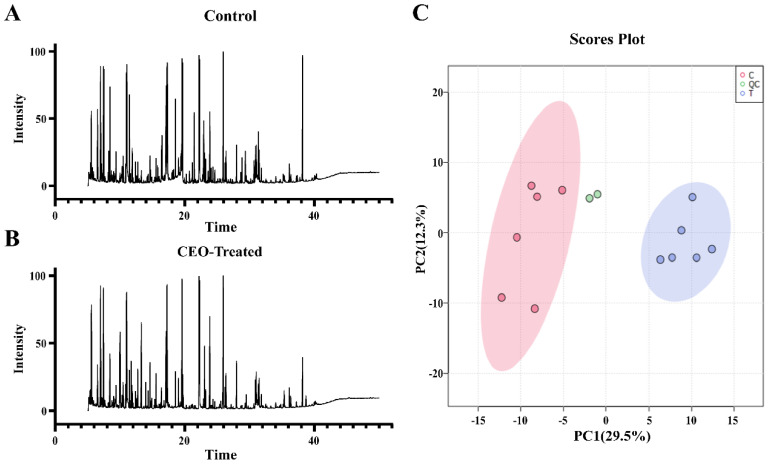
Metabolomic profiling of MRSA. (**A**): Representative total ion current chromatograms of a control sample. (**B**): Representative total ion current chromatograms of a CEO-treated sample. (**C**): PCA analysis of all samples.

**Figure 4 antibiotics-12-00591-f004:**
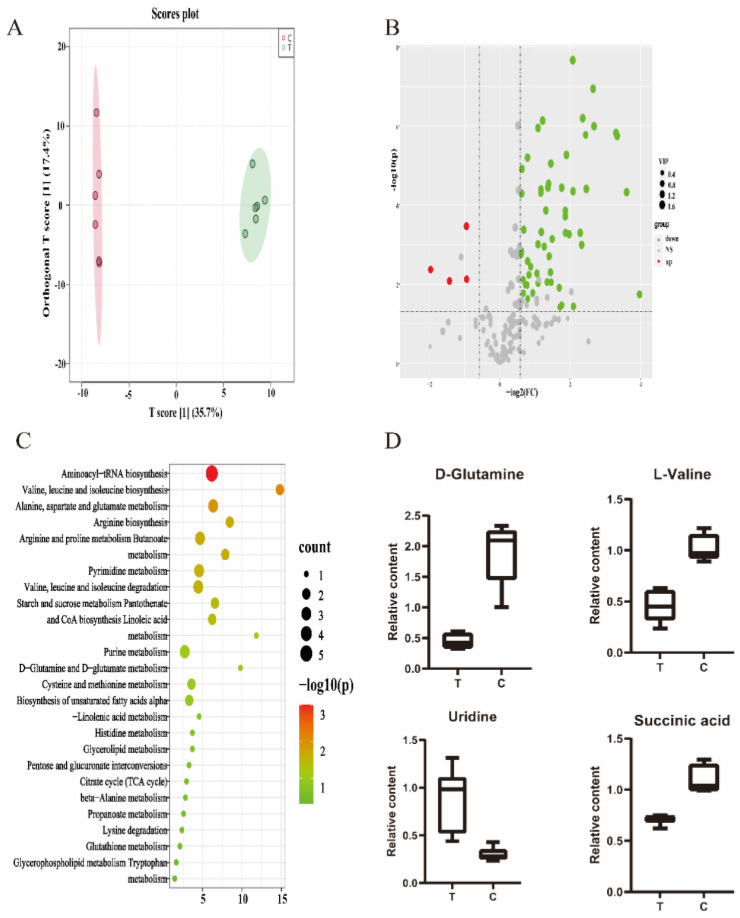
Data analysis of metabonomics. (**A**): OPLS-DA analysis. (**B**): Volcano map of all metabolites detected. (**C**): KEGG enrichment analysis. (**D**): Differences in the content of some differential metabolites between the control group and CEO-treated group.

**Table 1 antibiotics-12-00591-t001:** The main component of CEO.

No.	Component	RT	RI	Percentage (%)	Structure
1	*β*-Pinene	6.652	990	0.11	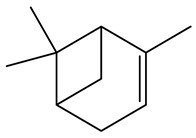
2	Limonene	7.989	1027	28.55	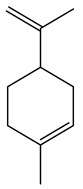
3	*γ*-Terpinene	9.146	1057	0.15	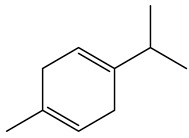
4	Dihydrocarvone	14.319	1194	0.13	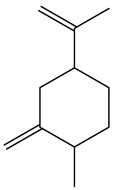
5	Dihydrocarvone	14.584	1201	0.10	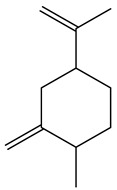
6	Carveol	15.161	1217	0.07	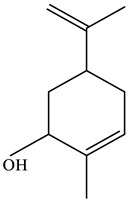
7	Ethanol, 2-(3,3-dimethylcyclohexylidene)	15.479	1226	0.11	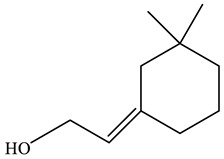
8	Carvone	16.039	1242	69.78	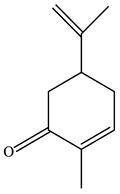
9	Perilla aldehyde	17.042	1271	0.14	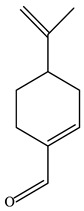
10	Others			0.85	
Total				100.00	

**Table 2 antibiotics-12-00591-t002:** Antibacterial activities of CEO.

Bacteria	Strains	CEO (*v*/*v*)		Chlorhexidine (μg/mL)
		MIC	MBC	MIC
MRSA	ATCC 43300	0.16%	0.64%	6.4
*S. aureus*	ATCC 6538	0.16%	0.32%	6.4
*E. coli*	ATCC 43895	0.16%	0.32%	6.4
*E. coli*	ATCC 25932	0.16%	0.32%	6.4

## Data Availability

Data will be made available on reasonable request.

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
