# Peer review of "Comprehensive Study of Components and Antimicrobial Properties of Essential Oil Extracted from Carum carvi L. Seeds"

_antibiotics, 2023, doi:10.3390/antibiotics12030591_

Round 1

Reviewer 1 Report

The study proposed by the authors presents an interesting area of health and medicine news, which causes interest for both scientists and practitioners.

The authors, no doubt, did a good job, including the application of modern methods and statistic studies in this research.

The first remark of the manuscript is lack of specific goal and too general introduction of the presented research.

While reading the article, another remark arised, by answering which the authors will improve the presentation of the results: line 228-229 (4. discussions) requires clarification

Author Response

Response to Reviewer 1 Comments

Point 1: The first remark of the manuscript is lack of specific goal and too general introduction of the presented research.

Response: We think this is an excellent suggestion, and have re-written this part.

Point 2: While reading the article, another remark arised, by answering which the authors will improve the presentation of the results: line 228-229 (4. discussions) requires clarification.

Response: We think this is an excellent suggestion.

Reviewer 2 Report

Additional comments:

Introduction

Many typo errors were found in the manuscript.

Material and methods

Line 101. Please include the meaning of MIC abbreviation.

Line 108. The concentrations (0.08%, 0.16%, 0.32%) used in time kill assay are fractions of the MIC? If it is correct, please include this in the text.

Line 111. More detail could be included in the crystal violet assay, for example the steps for mature biofilm inhibition. Please include also the concentration used.

Line 119. Why the concentration of 0.08% CEO was selected to evaluate the metabolite analysis?

Line 145.  Please check citation format.

Line 161. Please check the redaction.

Line 169. Please check English redaction in all manuscript.

In general, more methodological details are needed in this section.

Results and discussion

It is necessary a better description of the results.

It is necessary to improve the discussion of the results, comparing these with those reported for others authors.

Please include an explanation about the reported mechanism of action of CEO.

Author Response

Response to Reviewer 2 Comments

Point 1: Many typo errors were found in the manuscript.

Response: We apologize for the language of our manuscript. We have now worked on both language and readability and have also involved native English speaker for language corrections. We really hope that the language level have been substantially improved. The method section has also been supplemented and improved.

Point 2: Line 101. Please include the meaning of MIC abbreviation.

Response: We apologize for our carelessness, and we have added the meaning of MIC abbreviation.

Point 3: Line 108. The concentrations (0.08%, 0.16%, 0.32%) used in time kill assay are fractions of the MIC? If it is correct, please include this in the text.

Response: We apologize for our carelessness, and we have added it in the manuscript.

Point 4: Line 111. More detail could be included in the crystal violet assay, for example the steps for mature biofilm inhibition. Please include also the concentration used.

Response: We apologize for our carelessness, and the details of the experiment are added.

Point 5: Line 119. Why the concentration of 0.08% CEO was selected to evaluate the metabolite analysis?

Response: In our study, the MIC value of CEO against MRSA is 0.16%. CEO at the concentration of 0.08% could affect the growth of MRSA instead of killing them.

Point 6: Line 145. Please check citation format.

Response: We apologize for our carelessness, and it has been corrected.

Point 7: Line 161. Please check the redaction.

Response: We apologize for our carelessness, and it has been corrected.

Point 8: It is necessary a better description of the results.

Response: We are sorry about this and revised the manuscript.

Point 9: It is necessary to improve the discussion of the results, comparing these with those reported for others authors.

Response: We are sorry about this and revised the manuscript.

Point 10: Please include an explanation about the reported mechanism of action of CEO.

Response: We are sorry about this and revised the manuscript.

Reviewer 3 Report

In here the authors presented a metabolomic study on caraway essential oil against MRSA. The authors used a GC-Q-TOF-MS in untargeted mode to determine changes in the metabolomic profile of MRSA. Essential oil was characterized using the same GC-MS system. Concentration of terpenes were  determined using the normalization procedure. In general, the idea the evaluate the effect of terpenes on the metabolomic level is quite interesting. However, in our opinion the study is not well/correctly conducted. Especially the discovery and postulating tramadol (a synthetic opioid) as an important metabolic marker for the antibacterial effect of  caraway essential oil points towards a general misunderstanding on how to interpret MS data for metabolomic studies. Furthermore, there are some reservations on the chemometric analysis since the analysis is not correctly described. Incorrect performance of chemometric analysis can easily be misguiding the reader. Therefore we recommend rejection.

Detailed major and minor comments on the presented study can be find below, respectively.

General remarks:

-          major corrections concerning English grammar. I have not pointed out all the lines that need correction. More specific and non-misguiding wording should be used.

-          Interesting study which increasing importance, however flaws in the metabolomics analysis and no reliability in the identification of compounds

-          Quality assurance for metabolomics analysis is lacking. I highly recommend reading about mQACC (Metabolomics Quality Assurance & Quality Control)

-          Quantification of essential oils by normalization procedure highly depends on the used detector. We recommend an absolute quantification as a prerequisite for metabolomic studies and biological activity studies.

Abstract:

Line 18: singular (vegetable, food spice, preservative, herbal medicine)

Line 25: grammar – sentence is a bit confusion and should be restructured

Line 27/28: “potential for use in food and drug” – what do you mean by this? That you can use it as a drug or that you can add it do food/drugs as a preservative?

Introduction:

Line 36: English language: “And it would even lead to fatal bacteremia and sepsis when it comes to  more serious”. I’d suggest something line “In serious cases S. aureus infections can lead to fatal bacteremia and sepsis”.

Line 42: I don’t understand that sentence: “It was reported that the sensitivity of planktonic bacteria to antibiotics is about 10-1000 times that 44 of biofilms” – Do you mean you need 10 – 1000x as much antibiotics?

Materials and Methods:

Lines 74-80: When was Carum carvi collected (spring/summer, year)? What parts of the plant? Was it in bloom? Were all samples collected on the same day? This is very important, because natural products highly fluctuate in their contents depending on harvesting time, environment etc.

Line 87: Seeds were distillated – does this mean the plant was dried beforehand?

Line 88: for analysis did you use an internal standard for the CEO samples?

Line 91: quality of helium gas?

Line 108: How were the concentrations of CEO selected?

Line 129: GC-MS can be used for metabolomic studies, however some metabolites might not be particularly heat stable and can influence the outcome of the study. Interesting would further be a an LC-MS study. Analysis is good for apolar, heat stable metabolites. 

Line 133: odd selection of mass range

Line 138: was data further pre-processed? Mean centering/normalization procedures? If you process data like that for further chemometric analysis, some metabolites will overweight others just because of very high concentrations. You should try to see what happens, when all metabolites have the same weight in the analysis. Minor metabolites might be greatly overlooked, although important

Line 142: “Metabolites with VIP >1, fold change >1.5 or fold change  <0.67 were considered to be differential metabolites.” – I think you mean, that they appeared in significant concentrations. Further you did not mention about having any control groups for the study – this appears important because you are comparing cell cultures before and after a treatment.

Results:

Table 1: D-limonene is indicated however stereochemistry is not shown in the molecular structure. How can the authors be sure that it is the D-form. GC analysis on a non-chiral column (HP-5MS ) was performed. What about carvone?

151: concentration was determined probably comparing the areas? Can you imagine that that might just be a rough estimate? What are the SD for the concentrations?

155: table 1 – how stable were RT during analysis? 3 significant digits might be too high for reporting. Were MS spectra for compounds further compared to a library as a reference? In methods you described that identification was only conducted with RI. One dimension (in this case RI) is mostly not sufficient to definitely identify a compound.

191: which two groups? You never mentioned a control group. Furthermore, metabolomic studies contain a lot of variability. Differences can just be normal fluctuations or fluctuations in the measurement. How did you control for them? Usage of QCs during the measurement? Pooled samples?

Figure 3: You obviously further treated the data to produce such a PC plot by centering it/normalizing it, but you never mention it in your report. Was the analysis conducted with the same EO for all bacteria cultures?

Figure 4: D) why would the cells contain tramadol (an opioid drug) as a natural metabolite? I highly doubt that identity. Found tentatively identified metabolites should get their identity verified with reference compounds, retention times and MS spectra compared. B) is obviously a volcano and not a heat map.

Discussion:

Line 223: it highly depends how concentration was determined. I can imagine, that if you had used a calibration, concentration might have been different. You should have corrected injections with an internal standard.

Line 235: how do your results compare to usual treatments of MRSA? Can you use your CEO in concentrations comparable to antibiotics? A comparison would’ve been interesting for in-line treatment

Line 248: there are some terpenes, which are known to disrupt the cell membrane. Did you try to find information about this for limonene or carvone by themselves? Discussion is lacking, if single components could be used or if you find that the combination of different components might be important for antibacterial activity.

Author Response

Response to Reviewer 3 Comments

Point 1: Major corrections concerning English grammar. I have not pointed out all the lines that need correction. More specific and non-misguiding wording should be used.

Response: We apologize for the language of our manuscript. We have now worked on both language and readability and have also involved native English speaker for language corrections. We really hope that the language level have been substantially improved. The method section has also been supplemented and improved.

Point 2: Interesting study which increasing importance, however flaws in the metabolomics analysis and no reliability in the identification of compounds

Response: In the metabolomics analysis, the chemical compounds were discerned through the utilization of MassBank_NIST.msp in MSIDAL, wherein the screening threshold for identification was established to be greater than 60% similarity.

Point 3: Quality assurance for metabolomics analysis is lacking. I highly recommend reading about mQACC (Metabolomics Quality Assurance & Quality Control)

Response: Thanks for your advice.

Point 4: Quantification of essential oils by normalization procedure highly depends on the used detector. We recommend an absolute quantification as a prerequisite for metabolomic studies and biological activity studies.

Response: Thanks for your advice.

Point 5: Line 18: singular (vegetable, food spice, preservative, herbal medicine).

Response: We are sorry for this error and revised it.

Point 6: Line 25: grammar – sentence is a bit confusion and should be restructured.

Response: We are sorry for this error and revised it.

Point 7: Line 27/28: “potential for use in food and drug” – what do you mean by this? That you can use it as a drug or that you can add it do food/drugs as a preservative?

Response: Yes, the seeds of Carum carvi L is a traditional drug used for treating stomachache. And there were studies talk about the antibacterial of its essential oil.

Point 8: Line 36: English language: “And it would even lead to fatal bacteremia and sepsis when it comes to more serious”. I’d suggest something line “In serious cases S. aureus infections can lead to fatal bacteremia and sepsis”.

Response: Thank you for your advice, and it has been corrected.

Point 9: Line 42: I don’t understand that sentence: “It was reported that the sensitivity of planktonic bacteria to antibiotics is about 10-1000 times that 44 of biofilms” – Do you mean you need 10 – 1000x as much antibiotics?

Response: No, compared with planktonic bacteria, antibiotics need higher concentration to remove biofilm.

Point 10: Lines 74-80: When was Carum carvi collected (spring/summer, year)? What parts of the plant? Was it in bloom? Were all samples collected on the same day? This is very important, because natural products highly fluctuate in their contents depending on harvesting time, environment etc.

Response: Carum carvi was purchased from a commercial market in Xining,Qinghai province, China. As we know, all samples were collected on the same day.

Point 11: Line 87: Seeds were distillated – does this mean the plant was dried beforehand?

Response: Yes, it was conducted by air drying.

Point 12: Line 88: for analysis did you use an internal standard for the CEO samples?

Response: No, we used the peak area normalization method to detect its content.

Point 13: Line 91: quality of helium gas?

Response: We apologize for our carelessness, the quality of helium gas was 99.999% and it has been added in the manuscript.

Point 14: Line 108: How were the concentrations of CEO selected?

Response: In our study, the MIC value of CEO against MRSA is 0.16%. CEO at the concentration of 0.08% could affect the growth of MRSA instead of killing them.

Point 15: Line 129: GC-MS can be used for metabolomic studies, however some metabolites might not be particularly heat stable and can influence the outcome of the study. Interesting would further be a LC-MS study. Analysis is good for apolar, heat stable metabolites.

Response: Thank you for your advice, for metabolomic studies, LC-MS is better than GC-MS, but due to the limitation of laboratory conditions, we only did GC-MS to analyze some of the metabolites here.

Point 16: Line 133: odd selection of mass range

Response: We are sorry that we did not describe it clearly in the manuscript. The odd selection of mass range was 50~600 m/z.

Point 17: Line 138: was data further pre-processed? Mean centering/normalization procedures? If you process data like that for further chemometric analysis, some metabolites will overweight others just because of very high concentrations. You should try to see what happens, when all metabolites have the same weight in the analysis. Minor metabolites might be greatly overlooked, although important

Response: We are sorry that we did not describe it clearly in the manuscript. For the peak area obtained after deconvolution, we will correct the data based on the peak area of the internal standard and the weight of the sample. Subsequently, mean centering was applied.

Point 18: Line 142: “Metabolites with VIP >1, fold change >1.5 or fold change <0.67 were considered to be differential metabolites.” – I think you mean, that they appeared in significant concentrations. Further you did not mention about having any control groups for the study – this appears important because you are comparing cell cultures before and after a treatment.

Response: We are sorry that we did not describe it clearly in the manuscript. The control group is MRSA without CEO treatment.

Point 19: Table 1: D-limonene is indicated however stereochemistry is not shown in the molecular structure. How can the authors be sure that it is the D-form. GC analysis on a non-chiral column (HP-5MS) was performed. What about carvone?

Response: We are sorry for this mistake, HP-5MS column cannot determine the absolute configuration of the compound, and it has been revised.

Point 20: 151: concentration was determined probably comparing the areas? Can you imagine that that might just be a rough estimate? What are the SD for the concentrations?

Response: Yes, it is a simple estimate of the percentage of components in essential oil.

Point 21: 155: table 1 – how stable were RT during analysis? 3 significant digits might be too high for reporting. Were MS spectra for compounds further compared to a library as a reference? In methods you described that identification was only conducted with RI. One dimension (in this case RI) is mostly not sufficient to definitely identify a compound.

Response: After many times of analysis at different times, RT is stable. The identification was conduct by searching in NIST 14 library and the calculation of RI values.

Point 22: 191: which two groups? You never mentioned a control group. Furthermore, metabolomic studies contain a lot of variability. Differences can just be normal fluctuations or fluctuations in the measurement. How did you control for them? Usage of QCs during the measurement? Pooled samples?

Response: We are sorry that we did not describe it clearly in the manuscript. The control group is MRSA without CEO treatment. We try our best to ensure that the training conditions are exactly the same, and QC samples were used to evaluate the stability of instrument in the test. The QC samples were created by taking the same amount of each sample and pooling together.

Point 23: Figure 3: You obviously further treated the data to produce such a PC plot by centering it/normalizing it, but you never mention it in your report. Was the analysis conducted with the same EO for all bacteria cultures?

Response: We are sorry that we did not describe it clearly in the manuscript. PCA analysis was based on the data produced by the internal standard, the weight of the sample, and mean centering.

Point 24: Figure 4: D) why would the cells contain tramadol (an opioid drug) as a natural metabolite? I highly doubt that identity. Found tentatively identified metabolites should get their identity verified with reference compounds, retention times and MS spectra compared. B) is obviously a volcano and not a heat map.

Response: We are sorry for these mistakes, and the identification information of the compound and the name of Figure 4 B were corrected.

Point 25: Line 223: it highly depends how concentration was determined. I can imagine, that if you had used a calibration, concentration might have been different. You should have corrected injections with an internal standard.

Response: We agree that an internal standard will be helpful for determining the concentration. In fact, the peak area normalization method was used to detect relative content of compounds in essential oil in this study.

Point 26: Line 235: how do your results compare to usual treatments of MRSA? Can you use your CEO in concentrations comparable to antibiotics? A comparison would’ve been interesting for in-line treatment

Response: Compare to usual treatments of MRSA, there is still a certain gap in our results. But we think the advantage of CEO is that it is a natural product.

Point 27: Line 248: there are some terpenes, which are known to disrupt the cell membrane. Did you try to find information about this for limonene or carvone by themselves? Discussion is lacking, if single components could be used or if you find that the combination of different components might be important for antibacterial activity.

Response: Yes, it has been observed that limonene and carvone possess certain antibacterial properties, and that varying proportions of these ingredients can yield varying levels of antibacterial activity. Further research is being conducted to explore the potential of limonene and carvone, as well as other compounds, in this regard.

Reviewer 4 Report

The authors have described the antimicrobial and antibiofilm activity of essential oil extracted from Carum carvi L seeds. 

General comments:

Editing for the English language is required throughout the manuscript due to too many mistakes. The manuscript needs revision for language and grammar. The method section is poorly written especially the biological experiments-related part.

 Specific comments:

Line 102: The authors should mention the OD value.

Line 144: How long biofilm was grown? Did the authors confirm the maturation of biofilm?

The results for the antibiofilm activity should be given in terms of biomass and not in the OD.

Also, the authors should clearly define in the method at what stage the CEO was added to assess antibiofilm activity.

Further, the authors should perform a biofilm eradication assay to see the potential of CEO for the treatment of staphylococcal infections.  

The authors must perform a toxicity assay at least for the IC50 and IC90 CEO. The MIC value is only valid if that concentration is not toxic to the mammalian primary cell line.

Why only MRSA was chosen? Is the CEO selective towards MRSA? The authors should include a few common gram-positive and gram-negative pathogens to provide a comparative study.

The antibiotic control is missing throughout the experiment. The authors must use controls to validate their experiment.

How biofilm clearance was calculated? The authors should describe it either in methods or results.

How 0.08% CEO was calculated for the metabolite experiment? The authors need to discuss this.

There are a lot of statements that are either vague or have no meaning. For example:

In brief, the CEO in Xinjiang is similar to that in other places, but it is a bit different.” This statement is vague and has no meaning. The authors should provide a reasonable reason in the discussion why the concentration of limonene and carvone is different from previous studies.

“Herein, we focus on the anti-microorganisms activity of the CEO.” What is anti-microorganisms activity? It should be antimicrobial activity.

“Additionally, the impact on the biofilm of MRSA by CEO confirmed other authors’ results that CEO could destroy the biofilm by inhibiting the quorum sensing (QS)” The authors have not performed any experiment to analyze anti-virulence activity.

Author Response

Response to Reviewer 4 Comments

Point 1: Editing for the English language is required throughout the manuscript due to too many mistakes. The manuscript needs revision for language and grammar. The method section is poorly written especially the biological experiments-related part.

Response: We apologize for the language of our manuscript. We have now worked on both language and readability and have also involved native English speaker for language corrections. We really hope that the language level have been substantially improved. The method section has also been supplemented and improved.

Point 2: Line 102: The authors should mention the OD value.

Response: We are sorry that we did not describe it clearly in the manuscript. We observe its MCI by visual method.

Point 3: Line 144: How long biofilm was grown? Did the authors confirm the maturation of biofilm?

Response: We are sorry that we did not describe it clearly in the manuscript. The biofilm was grown for 24h. When the biofilm grows to 24 hours, the biomass tends to be stable.

Point 4: The results for the antibiofilm activity should be given in terms of biomass and not in the OD.

Response: We are sorry for this error and revised it.

Point 5: Also, the authors should clearly define in the method at what stage the CEO was added to assess antibiofilm activity.

Response: We apologize for our carelessness, and the details of the experiment are added.

Point 6: Further, the authors should perform a biofilm eradication assay to see the potential of CEO for the treatment of staphylococcal infections. 

Response: We are sorry that we did not describe it clearly in the manuscript. The biofilm eradication was analysis by the clearance of mature biofilm.

Point 7: The authors must perform a toxicity assay at least for the IC50 and IC90 CEO. The MIC value is only valid if that concentration is not toxic to the mammalian primary cell line.

Response: We agree that, and the further study will focus on the toxicity and antibacterial of CEO at the cellular and in vivo.

Point 8: Why only MRSA was chosen? Is the CEO selective towards MRSA? The authors should include a few common gram-positive and gram-negative pathogens to provide a comparative study.

Response: We are sorry that we did not describe it clearly in the manuscript. In this study, four bacteria were chosen and the antibacterial activity of CEO against them is similar. In view of this result, we chose MRSA for further study.

Point 9: The antibiotic control is missing throughout the experiment. The authors must use controls to validate their experiment.

Response: We apologize for our carelessness, and the antibiotic control has been added in to the manuscript.

Point 10: How biofilm clearance was calculated? The authors should describe it either in methods or results.

Response: We apologize for our carelessness. The biofilm clearance was calculated by the following formula, and it has been added in to the manuscript.

Point 11: How 0.08% CEO was calculated for the metabolite experiment? The authors need to discuss this.

Response: In our study, the MIC value of CEO against MRSA is 0.16%. CEO at the concentration of 0.08% could affect the growth of MRSA instead of killing them.

Point 12: There are a lot of statements that are either vague or have no meaning. For example:

“In brief, the CEO in Xinjiang is similar to that in other places, but it is a bit different.” This statement is vague and has no meaning. The authors should provide a reasonable reason in the discussion why the concentration of limonene and carvone is different from previous studies.

“Herein, we focus on the anti-microorganisms activity of the CEO.” What is anti-microorganisms activity? It should be antimicrobial activity.

Response: We apologize for the poor language of our manuscript. And the manuscript was examined and corrected carefully.

Point 13: “Additionally, the impact on the biofilm of MRSA by CEO confirmed other authors’ results that CEO could destroy the biofilm by inhibiting the quorum sensing (QS)” The authors have not performed any experiment to analyze anti-virulence activity.

Response: We are sorry that we did not describe it clearly in the manuscript and it has been corrected.

Round 2

Reviewer 2 Report

The suggested corrections was included and the quality and content was improved.

Reviewer 3 Report

Ready for publishing now